# Persistence of T-Cell Immunity Responses against SARS-CoV-2 for over 12 Months Post COVID-19 Infection in Unvaccinated Individuals with No Detectable IgG Antibodies

**DOI:** 10.3390/vaccines11121764

**Published:** 2023-11-27

**Authors:** Vassiliki C. Pitiriga, Myrto Papamentzelopoulou, Kanella E. Konstantinakou, Irene V. Vasileiou, Konstantina S. Sakellariou, Natalia I. Spyrou, Athanasios Tsakris

**Affiliations:** 1Department of Microbiology, Medical School, National and Kapodistrian University of Athens, 75 Mikras Asias Street, 11527 Athens, Greece; atsakris@med.uoa.gr; 2Molecular Biology Unit, 1st Department of Obstetrics and Gynecology, National and Kapodistrian University of Athens, 11528 Athens, Greece; mpntua@yahoo.gr; 3Bioiatriki Healthcare Group, Kifisias 132 and Papada Street, 11526 Athens, Greece; nkonstantinakou@bioiatriki.gr (K.E.K.); irenebasileiou@gmail.com (I.V.V.); konstanssakellariou@gmail.com (K.S.S.); nataspyrou@gmail.com (N.I.S.)

**Keywords:** cellular immunity, T-cell immunity, SARS-CoV-2, COVID-19, coronavirus, vaccination, natural infection, IFN-γ, ELISpot, IGRA

## Abstract

Background: Immune response to SARS-CoV-2 is crucial for preventing reinfection or reducing disease severity. T-cells’ long-term protection, elicited either by COVID-19 vaccines or natural infection, has been extensively studied thus far; however, it is still attracting considerable scientific interest. The aim of the present epidemiological study was to define the levels of T-cellular immunity response in a specific group of unvaccinated individuals from the general population with a prior confirmed COVID-19 infection and no measurable levels of IgG antibodies. Methods: We performed a retrospective descriptive analysis of data collected from the medical records of consecutive unvaccinated individuals recovered from COVID-19, who had proceeded to a large private medical center in the Attica region from September 2021 to September 2022 in order to be examined on their own initiative for SARS-CoV-2 T-cell immunity response. The analysis of T-cell responses was divided into three time periods post infection: Group A: up to 6 months; Group B: 6–12 months; Group C: >12 months. The SARS-CoV-2 T-cell response was estimated against spike (S) and nucleocapsid (N) structural proteins by performing the T-SPOT. COVID test methodology. SARS-CoV-2 IgG antibody levels were measured by the SARS-CoV-2 IgG II Quant assay (Abbott Diagnostics). Results: A total of 182 subjects were retrospectively included in the study, 85 females (46.7%) and 97 (53.3%) males, ranging from 19 to 91 years old (mean 50.84 ± 17.2 years). Among them, 59 (32.4%) had been infected within the previous 6 months from the examination date (Group A), 69 (37.9%) had been infected within a time period > 6 months and <1 year (Group B) and 54 (29.7%) had been infected within a time period longer than 1 year from the examination date (Group C). Among the three groups, a positive T-cell reaction against the S antigen was reported in 47/58 (81%) of Group A, 61/69 (88.4%) of Group B and 40/54 (74.1%) of Group C (chi square, *p* = 0.27). T-cell reaction against the N antigen was present in 45/58 (77.6%) of Group A, 61/69 (88.4%) of Group B and 36/54 (66.7%) of Group C (chi square, *p* = 0.02). The median Spot-Forming Cells (SFC) count for the S antigen was 18 (range from 0–160) in Group A, 19 (range from 0–130) in Group B and 17 (range from 0–160) in Group C (Kruskal–Wallis test, *p* = 0.11; pairwise comparisons: groups A–B, *p* = 0.95; groups A–C, *p* = 0.89; groups B–C, *p* = 0.11). The median SFCs count for the N antigen was 14.5 (ranging from 0 to 116) for Group A, 24 (ranging from 0–168) in Group B and 16 (ranging from 0–112) for Group C (Kruskal–Wallis test, *p* = 0.01; pairwise comparisons: groups A–B, *p* = 0.02; groups A–C, *p* = 0.97; groups B–C, *p* = 0.03). Conclusions: Our data suggest that protective adaptive T-cellular immunity following natural infection by SARS-CoV-2 may persist for over 12 months, despite the undetectable humoral element.

## 1. Introduction

One of the most crucial coronavirus disease 2019 (COVID-19) pandemic research fields involves the components of both humoral and cellular immunity induced after vaccination or a natural infection by severe acute respiratory syndrome coronavirus 2 (SARS-CoV-2) in a longitudinal manner [1]. With growing knowledge in this field, the effectiveness of public health guidelines can be further enhanced. As known so far, specific adaptive immune response is developed both after natural infection and vaccination to control and eliminate SARS-CoV-2 infection [2]. In terms of vaccination, a significant number of the vaccines that have been developed up to now exhibit high levels of protection, with particularly noticeable efficacy against reinfection and serious disease [3]. Similarly, it is well documented that in natural COVID-19 infection the presence of clearance-neutralizing antibodies and antigen-specific T-cells provide protection against both severe disease and death, while cellular immunity persists longer in time compared to humoral [3,4]. It is supported that the differentiation of stem cell memory T-cells (TSCMs) into various T-cell memory subsets may be a critical key step to developing robust protection against SARS-CoV-2 in individuals with prior COVID-19 infection [5]. It still remains crucial to fully understand the role of cellular immunity in the long-term protection against SARS-CoV-2 in order to define the mechanism of the protective adaptive responses, especially in previously infected individuals exhibiting different clinical severity levels [6,7].

Currently, the broad implementation of advanced laboratory technologies, including molecular and cellular assays, improves T-cell response estimation, thus providing meticulous T-cell response analysis. Among those, interferon-gamma release assays (IGRAs), functional cellular assays based on the detection of interferon-gamma (IFN-γ), have proven particularly useful [8,9]. As demonstrated, the IFN-γ T-cell responses against SARS-CoV-2 are generally low during the acute phase of the infection but increase in COVID-19-convalescent individuals [10].

In this field, major concern rises for the longevity and profile of adaptive natural immunity in unvaccinated COVID-19-convalescent individuals. Several studies have disclosed that T-cell response to SARS-CoV-2 and humoral natural immunity may persist for a period of more than one year after infection [11,12,13,14,15]. Recent studies have demonstrated the presence of highly immunogenic anti-spike and anti-nucleocapside antibodies along with robust memory T-cell responses in unvaccinated convalescent subjects, with an estimated longevity range between 6 and 18 months [16,17,18,19,20]. As for convalescent subjects exhibiting severe COVID-19, a robust memory immune response against SARS-CoV-2 was generated and maintained for more than one year after hospitalization [21].

Cellular responses are documented to persist even when the humoral immunity has waned or is no longer detectable [22]. As disclosed, SARS-CoV-2-specific T-cells were detectable in IgG-negative convalescent individuals with a history of asymptomatic or mild COVID-19 [3,4]. Another study provided evidence that seronegative COVID-19-convalescent individuals had developed a SARS-CoV-2 T-cell response comparable to seropositive ones [23]. As for the reinfection risk, recent longitudinal studies indicated that the presence of cellular and humoral immunity months after the initial infection could possibly decrease the risk of reinfection in the case of a new emerging SARS-CoV-2 variant or even protect from severe disease in case of reinfection [24,25,26].

Considering that the risk of COVID-19 reinfection remains a research field of interest, profiling of developed SARS-CoV-2 natural immunity is of high priority. Especially the knowledge on the waning patterns of human cellular immune responses after natural COVID-19 infection and its longevity in the general population after humoral element attenuation has not been thoroughly examined, while still arousing great research interest. Accordingly, the present epidemiological retrospective study aims at estimating the levels and duration of T-cell immunity in unvaccinated COVID-19-convalescent individuals of the general population with no measurable levels of IgG antibodies.

## 2. Methods

### 2.1. Study Design

A retrospective data analysis was performed based on the electronic medical records of consecutive adult individuals who had proceeded to “BIOIATRIKI” healthcare center, a large medical center in the region of Attica, from September 2021 to September 2022, in order to be examined for COVID-19 cellular immunity response as part of a SARS-CoV-2 immunity screening during the pandemic. We included in the study unvaccinated COVID-19-convalescent subjects with a negative SARS-CoV-2 IgG antibody test on the examination day. The date of the first positive PCR result was considered the date of COVID-19 infection diagnosis. In cases where PCR was carried out as a confirmation test after a positive self-test of rapid test, the date of the first positive laboratory test was considered as the date of COVID-19 diagnosis. Individuals with a history of confirmed reinfections and immunocompromised patients were excluded from the study. The analysis of T-cell responses was divided into 3 time periods post infection: Group A: up to 6 months; Group B: 6–12 months; Group C: >12 months. Additional data in terms of clinical symptoms and medical history was obtained by structured questionnaires that were routinely filled by individuals at the time of examination. No records of clinical data such as chest X-rays and hospitalization records were obtained.

The study was approved by the institutional review board (date of approval: 29 June 2021, 6th annual meeting).

### 2.2. Laboratory Methods

#### 2.2.1. Enzyme-Linked Immunosorbent Spot (ELISpot) Assay for IFN-g T-Cell Response Detection

The T-cell response to COVID 19 infection was assessed by the IGRA methodology T-SPOT^®^.COVID (Oxford Immunotec, Oxfordshire, UK). The T-SPOT.COVID test is a standardized ELISpot-based method for recognition of T-cell immune response to SARS-CoV-2 in samples of whole blood. The test uses the established T-SPOT technology with an antigen mix that includes over 250 SARS-CoV-2 peptides in 2 antigen peptide pools; one pool contained peptides originating from the spike S1 protein and the other contained peptides from the N nucleocapsid protein.

Blood samples were handled according to the manufacturer’s instructions. After adding the T-cell Xtend^®^ reagent (Oxford Immunotec, UK) to the sample tubes, the peripheral blood mononuclear cells were isolated by density gradient centrifugation from a whole blood sample, washed and then counted before being added into the test. An approximate number of 250,000 cells/well were plated into 4 wells of a 96-well plate.

The two pools of antigen peptide were added to the two antigen wells; the T-cell mitogen phytohemagglutinin was added to the positive control well, and cell culture media alone was added to the negative control well. The wells were washed after 16–20 h of incubation, and a conjugated secondary antibody with the ability to connect to any IFN-γ captured on the membrane was added. Over the course of the procedure, the wells were washed to remove the IFN-γ that was not connected, and a substrate was added to produce the typical dark color dots of products demonstrating regions of IFN-γ presence. The spot-forming cells (SFCs) were manually counted by microscopy by medical doctors and experienced technologists. Results were reported for N and S antigens separately and were expressed as ‘invalid’ if the negative control had more than 10 SFCs or the positive control had fewer than 20 SFCs when the antigen wells were non-reactive. The method cut-off was predefined at 6 SFCs for each antigen. Based on the existing data, we introduced a borderline zone of +/−1 SFCs to account for potentially elevated test variability around the cut-off [27]. Accordingly, results were reported as S and/or N ‘reactive’, ‘non-reactive’ and ‘borderline’ when the SFCs in the antigens wells minus the SFCs of the negative control were ≥8, ≤4 and 5–7, respectively.

#### 2.2.2. SARS-CoV-2 IgG Antibodies

SARS-CoV-2 IgG antibody levels were measured in participants’ blood sera the same day of examination using the SARS-CoV-2 IgG II Quant assay from Abbott Laboratories (quantitative method). This method is an automated chemiluminescent microparticle immunoassay (CMIA) that is carried out for the qualitative and quantitative determination of IgG antibodies specific to the receptor-binding domain (RBD) of SARS-CoV-2. The sequence used for the RBD was taken from the WH-Human 1 coronavirus, GenBank accession number MN908947. The analytical measurement interval is stated by the manufacturer as 21–40,000 arbitrary units (AU)/mL and reported positivity cut-off is ≥50 AU/mL.

### 2.3. Statistical Analysis

We performed the chi-square (*χ*^2^) test to compare categorical variables and one-way ANOVA to compare continuous variables. The Kruskal–Wallis test was carried out in cases of non-parametric continuous variables. The Spearman’s rank correlation test was used to examine possible correlations between SFC counts of S and N antigens with the time interval after COVID-19 infection. The kinetics of SFC counts of S and N antigens over time are exhibited through the plotting charts of the curve estimation for linear regression models.

The statistical analysis was carried out using SPSS version 28 (IBM SPSS Statistics). Results were considered statistically significant when *p*-value was <0.05.

## 3. Results

### 3.1. Participants’ Demographical and Clinical Characteristics

Among the 182 total individuals who were chosen to enter the study, 84 were females (46.2%) and 97 (53.6%) were males, ranging from 19 to 91 years old (mean: 50.84 ± 17.2 years). The study participants’ characteristics are presented in Table 1. The SARS-CoV-2 IgG antibody testing demonstrated for all participants lower antibody levels than the positivity cut-off (mean value: 27.7 AU/mL).

Their mean time period post SARS-CoV-2 infection was 282.26 ± 175.55 days (ranging from 4–780 days). Among them, 59 (32.4%) had been infected within the previous 6 months from the examination date (Group A), 69 (37.9%) had been infected within a time period > 6 months and <1 year (Group B) and 54 (29.7%) had been infected in a time period longer than 1 year from the examination date (Group C).

No differences in proportion of sex and age were observed among the three groups. More specifically, in Group A, the proportion of sex was 33 females (55.9%) and 26 males (44.1%), in Group B it was 28 females (40.6%) and 41 males (59.4%) and in Group C it was 24 females (44.4%) and 30 males (55.6%) (*χ*^2^ test, *p* = 0.24). The mean age in Group A was 50.53 ± 15.75 years, in Group B it was 51.97 ± 17.67 years and in Group C it was 49.72 ± 18.48 (one-way ANOVA test, F = 0.269, *p* = 0.76).

### 3.2. Positivity Rate among Total Participants and between Groups

Of the total participants, a positive T-cell reaction to SARS-CoV-2 was exhibited in 148/182 (81.3%) against the S antigen and 142/182 (78.0%) against the N antigen. Both S and N antigens elicited T-cell responses in 139/182 (76.37%) cases. Borderline S results were reported in 6/182 (3.3%) and non-reactive in 28/182 (15.3%), while borderline N results were reported in 9/182 (5%) and non-reactive in 31/182 (17%). Four samples exhibited fewer than 20 SFCs in the positive control; however, two of them had a response against the S and the other two against N antigens. For this reason, these samples were considered positive only for the particular antigen.

Among the three groups, a positive reaction against the S antigen was reported in 47/58 (81%) of Group A, 61/69 (88.4%) of Group B and 40/54 (74.1%) of Group C (chi square, *p* = 0.27). Reaction against the N antigen was present in 45/58 (77.6%) of Group A, 61/69 (88.4%) of Group B and 36/54 (66.7%) of Group C (chi square, *p* = 0.02) (Figure 1a,b).

### 3.3. Quantitative T-Cell Response against SARS-CoV-2 Antigens

Among the total participants, the median SFC count for the S antigen was 18 (range from 0–160) and for the N antigen was 18 (range from 0–168). Distribution of T-SPOT results against S and N antigens is presented in Figure 2a,b. No difference was established in the median SFC count for the S antigen among the three groups. More specifically, the median SFC was 18 (range from 0–160) in Group A, 19 (range from 0–130) in Group B and 17 (range from 0–160) in Group C (Kruskal–Wallis test, *p* = 0.11; pairwise comparisons: groups A–B, *p* = 0.95; groups A–C, *p* = 0.89; groups B–C, *p* = 0.11). Regarding the N antigen, Group B exhibited significantly higher SFC counts compared to the other two groups. More specifically, the median SFC count for the N antigen was 14.5 (ranging from 0 to 116) for Group A, 24 (ranging from 0–168) in Group B and 16 (ranging from 0–112) for Group C (Kruskal–Wallis test, *p* = 0.01; pairwise comparisons: groups A–B, *p* = 0.02; groups A–C, *p* = 0.97; groups B–C, *p* = 0.03) (Figure 3).

### 3.4. T-Cell Response According to Days after Infection

In total participants, no statistically significant positive or negative correlations between the IFN-γ response to either SARS-CoV-2 S or N antigens and time after infection were established (SR = −0.028; *p* = 0.70 for S, SR = 0.02; *p* = 0.72 for N). These results suggest that S and N antigen levels remain stable over the specific time (Figure 4a,b).

The mean time period post infection for the three groups was as follows: Group A: 97.64 ± 60.44 days (range 4–210), Group B: 264.77 ± 52.75 days (range 181–365) and Group C: 506.31 ± 90.54 (range 369–780) days.

In Group A, a weak positive correlation between the T-cell response to S and N antigens and time after infection was demonstrated, indicating a rise in S and N antigen levels overtime, but not in a statistically significant level (Spearman’s rank correlation test, SR = 0.165; *p* = 0.21 for S, SR = 0.199; *p* = 0.13 for N). In Group B, no significant correlation between the IFN-γ response to N (Spearman’s rank correlation test, SR = 0.048; *p* = 0.69) and S (SR = −0.001; *p* = 0.99) antigens and time after infection was established. In Group C, there were no clear positive or negative trends observed between T-cell responses to S and N antigens and time interval since infection (Spearman’s rank correlation test, SR = 0.187; *p* = 0.17 for S, SR = −0.06; *p* = 0.65 for N). Kinetics over time (in days) for S and N antigens in subjects of Groups A, B and C are presented in Figure 5, Figure 6 and Figure 7 which exhibit the plotting charts of the curve estimation for linear regression models.

## 4. Discussion

Understanding the long-term adaptive humoral and cellular responses in the unvaccinated COVID-19-convalescent population is an essential key in estimating the longevity of the developed natural SARS-CoV-2 immunity and the potential risk and severity of reinfection. Moreover, comprehensively depicting the kinetics of protective cellular responses is a feasible approach to provide evidence for strengthening the effectiveness of COVID-19 vaccines by emphasizing the necessity of dual immunization to stimulate humoral and especially cellular responses. As demonstrated in the present retrospective study, the vast majority of the unvaccinated COVID-19-convalescent participants had persistent T-cell immunity, with no measurable levels of antibodies. In almost one third of the study participants, the levels of immune responses were detectable after at least 12 months. More specifically, the analysis of T-cell responses revealed that both S and N antigens were present for at least 12 months with rather comparable levels, while the highest SFC values were observed in Group B, between 6 and 12 months post infection. Accordingly, our results suggest that adaptive T-cellular immunity following COVID-19 infection maintains for at least one year, even in the absence of humoral immunity.

Our results are in line with several studies demonstrating that humoral responses wane over time, while T-cell immunity persists in unvaccinated COVID-19-convalescent individuals. In particular, one of the first studies that investigated the functional and phenotypic map of SARS-CoV-2-specific T-cell immunity in unvaccinated convalescent individuals with a history of mild or absence of symptoms revealed that robust and durable memory T-cell responses were detectable, even when humoral responses were absent almost two months after disease onset [1]. An early study showed a considerable decline of the anti-SARS-CoV-2 IgG antibodies after the 6th month post infection with the SARS-CoV-2-specific T- and/or memory B-cell responses persisting for 8 months after disease onset [28]. Another longitudinal analysis showed that humoral SARS-CoV-2-specific immune response declined over time, whereas the SARS-CoV-2 T-cell repertoire was stable for up to 102 days after symptoms onset [29]. Jung et al. reported that SARS-CoV-2-specific memory T-cell responses persisted in COVID-19-convalescent individuals 10 months post infection regardless of the disease severity, whereas SARS-CoV-2-specific antibody levels decreased [30]. In a recent longitudinal analysis, convalescent individuals with low/negative antibody titers presented a detectable level of SARS-CoV-2-specific T-cell response 9 months after mild COVID-19 disease onset [31]. Two independent research groups observed that SARS-CoV-2 memory B cells, CD8+ T cells, and CD4+ T cells remained measurable for more than 6 months after infection, whereas antibodies against SARS-CoV-2 spike and RBD waned gradually 8 months after COVID-19 onset [17,19].

Other studies present discrepant results, since they demonstrate the co-existence of humoral and cellular responses in unvaccinated COVID-19-convalescent individuals, at least in the first months post SARS-CoV-2 infection. It was recently demonstrated that both neutralizing antibodies and memory T-cell responses persisted up to 18 months in unvaccinated COVID-19-convalescent individuals, indicating a slow waning of immunity and, thus, an enhanced protection against reinfection [16]. Importantly, a recent systematic review and meta-analysis disclosed that protection against reinfection and severe disease conferred by both humoral and cellular immunity is maintained at a considerable level up to 12 months after initial SARS-CoV-2 infection [32]. A retrospective study disclosed that RBD-IgG, full-length spike-IgG and serum neutralizing capacity remained measurable for up to 1 year with persistent SARS-CoV-2-specific T-cell responses between 6 months and 12 months [33]. In a longitudinal cohort study, both humoral and cellular immunity against SARS-CoV-2 was detected in most convalescent individuals 12 months post moderate-to-critical infection; however, neutralizing antibody titers declined faster between 6 and 12 months post COVID-19 onset than SARS-CoV-2 cellular immunity [34]. In other longitudinal studies, the majority of convalescent individuals had measurable SARS-CoV-2 anti-spike IgG at least 3 [35] and 4 months [36] post infection, respectively, with persistent memory B and memory T cells.

Our study bears specific limitations, including those of an observational retrospective study. In this context, no valuable clinical and microbiological information for further analysis was available such as the different viral variants, the severity of the clinical symptoms and the need for hospitalization in order to classify our participants as mild, moderate and severe COVID-19 cases. Another limitation is the fact that no data was available regarding the kinetics over time of humoral immunity in our study group. Also, we should take into account the possibility that, even though we excluded individuals with confirmed COVID-19 reinfections, a portion of the participants could have had a history of an undiagnosed asymptomatic reinfection.

## 5. Conclusions

Our results increase the evidence of the presence of long-term adaptive cellular immunity in unvaccinated COVID-19-convalescent individuals. Broad application of both existing and innovative methods for measuring T-cell immunity along with a continuous follow-up of the convalescent population many years after SARS-CoV-2 infection could provide a more comprehensive kinetics profile of adaptive immune response, thus updating public health strategies.

## Figures and Tables

**Figure 1 vaccines-11-01764-f001:**
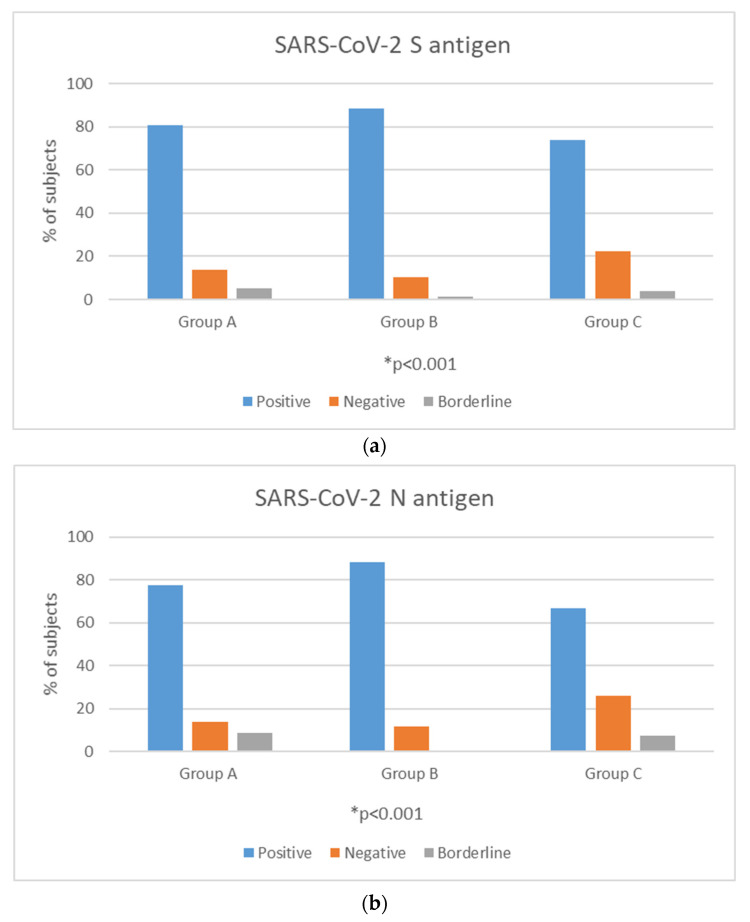
Positivity rates (%) of T-cell response against SARS-CoV-2 S (**a**) and N (**b**) antigens among time interval groups. ** p*-value < 0.05 is considered to be statistically significant.

**Figure 2 vaccines-11-01764-f002:**
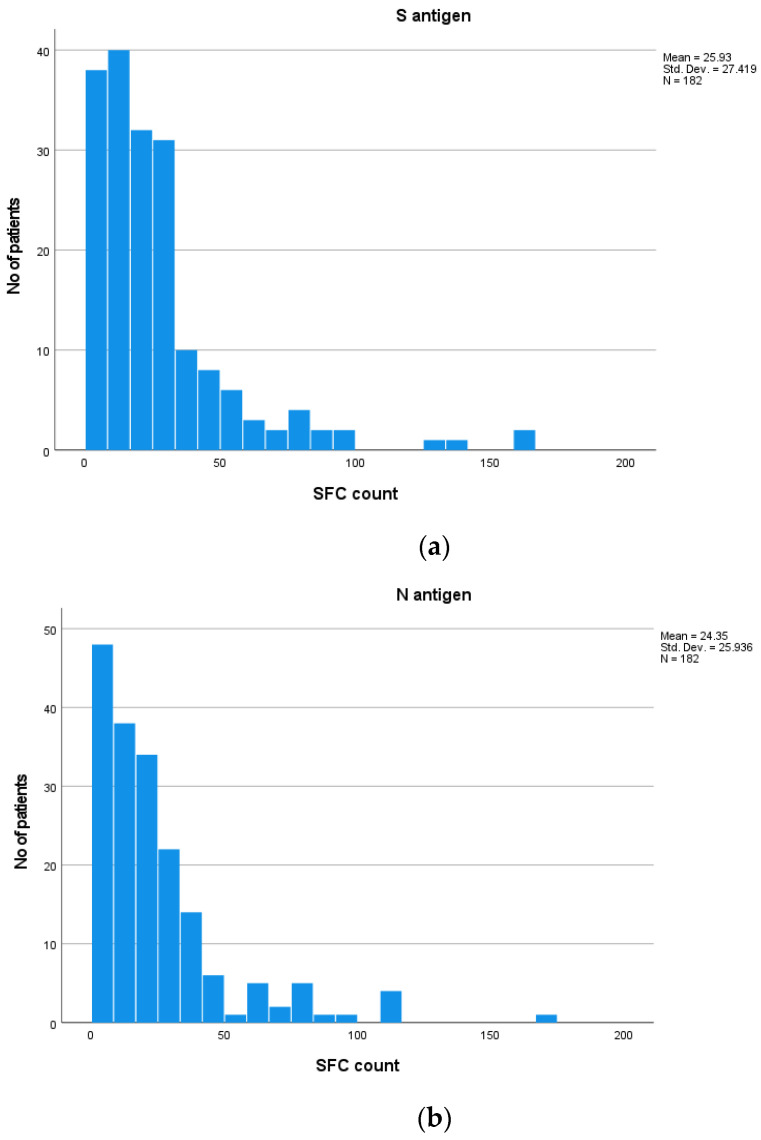
Distribution of T-SPOT results against S (**a**) and N (**b**) antigens among total participants.

**Figure 3 vaccines-11-01764-f003:**
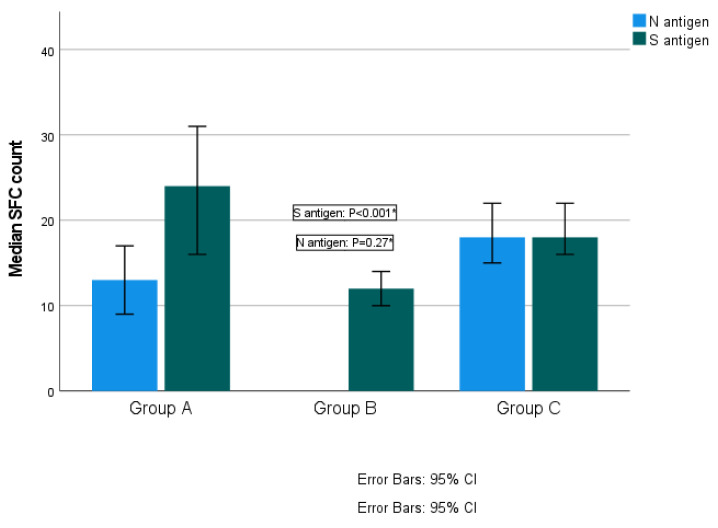
Quantitative T-SPOT results for T-cell responses against SARS-CoV-2 N and S antigens among time interval groups (*p*-values by Kruskal–Wallis test). * *p*-value < 0.05 is considered to be statistically significant.

**Figure 4 vaccines-11-01764-f004:**
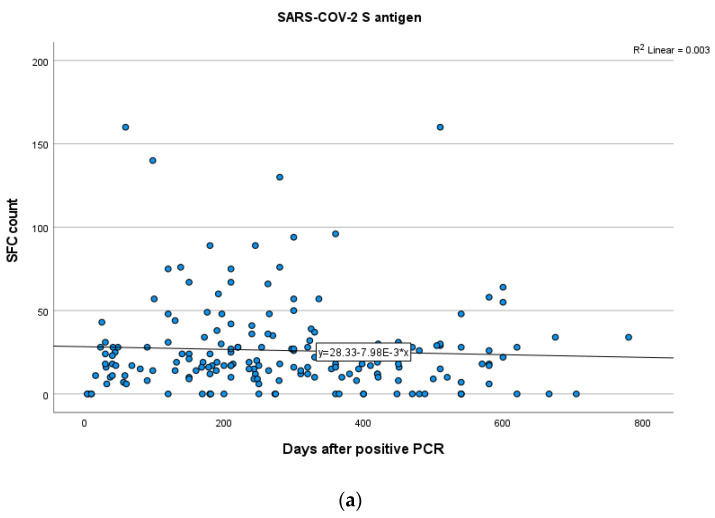
Kinetics over time (in days) of the quantitative T-SPOT results (dots in figures) for S (**a**) and N (**b**) antigens in total participants (curve estimation test for linear regression model).

**Figure 5 vaccines-11-01764-f005:**
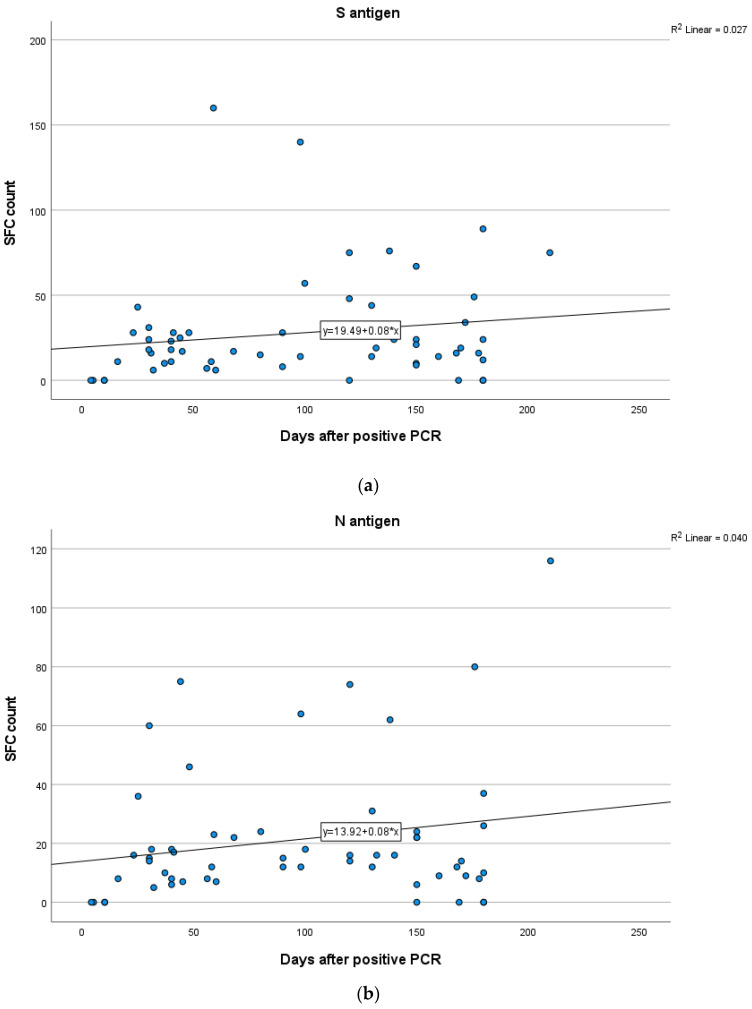
Kinetics over time (in days) of the quantitative T-SPOT results (dots in figures) for S (**a**) and N (**b**) antigens in Group A (curve estimation test for linear regression model).

**Figure 6 vaccines-11-01764-f006:**
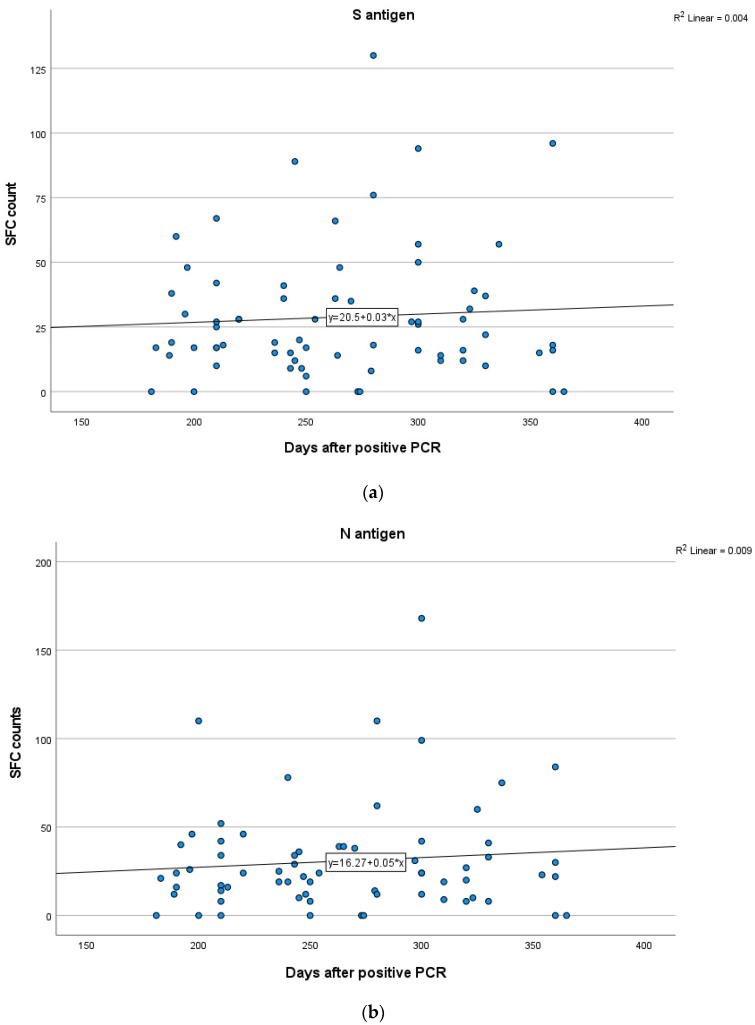
Kinetics over time (in days) of the quantitative T-SPOT results (dots in figures) for S (**a**) and N (**b**) antigens in Group B (curve estimation test for linear regression model).

**Figure 7 vaccines-11-01764-f007:**
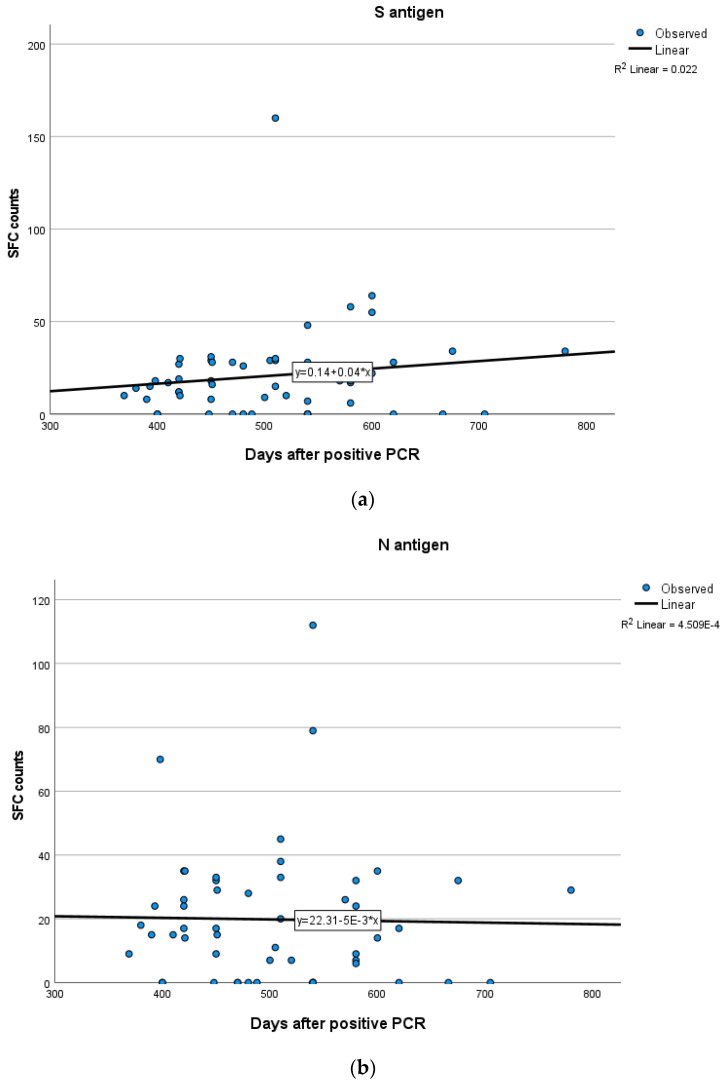
Kinetics over time (in days) of the quantitative T-SPOT results (dots in figures) for S (**a**) and N (**b**) antigens in Group C (curve estimation test for linear regression model).

**Table 1 vaccines-11-01764-t001:** Participants’ demographic characteristics and data of clinical symptoms and comorbidities. Groups are defined as follows. Group A: up to 6 months; Group B: 6–12 months; Group C: >12 months. * *p* < 0.05 is considered to be statistically significant. Abbreviations: NS, not significant; N, number of subjects.

	Group A (N = 59)	Group B (N = 69)	Group C (N = 54)	* *p*-Value
**Demographic Characteristics**				
Sex (F/M)	33/26	28/41	24/30	NS
Age (years)	50.53 ± 15.75	51.97 ± 17.67	49.72 ± 18.48	NS
Time period post infection (days)	97.64 ± 60.44	264.77 ± 52.75	506.31 ± 90.54	
**COVID-19 Clinical symptoms**				
Asymptomatic, n (%)	5 (8.4)	10 (14.5)	7 (12.9)	NS
Symptomatic, n (%)	54 (91.5)	59 (85.5)	47 (87.0)	NS
Shortness of breath, n (%)	9 (15.2)	8 (11.6)	4 (7.4)	NS
Sore throat, n (%)	50 (84.7)	40 (57.9)	45 (83.3)	NS
Fatigue, n (%)	54 (91.5)	5 (7.2)	41 (75.9)	NS
Loss of taste/smell, n (%)	12 (20.3)	20 (28.9)	11 (20.3)	NS
Diarrhea, n (%)	9 (15.2)	8 (11.6)	4 (7.4)	NS
Headache, n (%)	30 (50.8)	25 (36.2)	34 (62.9)	NS
Vomiting, n (%)	3 (5.0)	2 (2.9)	4 (7.4)	NS
Fever, n (%)	45 (76.2)	48 (69.5)	37 (68.5)	NS
**Comorbidities**				
Respiratory disorders n (%)	11 (18.6)	5 (7.2)	8 (14.8)	NS
Cardiovascular diseases n (%)	10 (16.9)	9 (13.0)	7 (12.9)	NS
Autoimmune disorders n (%)	10 (16.9)	12 (17.4)	7 (12.9)	NS
Central nervous system disorders n (%)	1 (1.7)	1 (1.4)	2 (3.7)	NS
Diabetes mellitus n (%)	5 (8.4)	4 (5.8)	2 (3.7)	NS
Hypertension n (%)	10 (16.9)	15 (21.7)	8 (14.8)	NS
Lipidemia n (%)	17 (28.8)	16 (23.1)	12 (22.2)	NS
Obesity n (%)	9 (15.2)	13 (18.8)	12 (22.2)	NS

## Data Availability

All data of this study are included in this article.

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
