# Peer review of "Persistence of T-Cell Immunity Responses against SARS-CoV-2 for over 12 Months Post COVID-19 Infection in Unvaccinated Individuals with No Detectable IgG Antibodies"

_vaccines, 2023, doi:10.3390/vaccines11121764_

Round 1
Reviewer 1 Report
Comments and Suggestions for Authors
This study focuses on assessing T cell immunity responses in individuals from the general population with a prior confirmed COVID-19 infection but no measurable levels of IgG antibodies. The research spans three time periods post-infection, and T-cell responses are evaluated using the T-SPOT.COVID test. The study suggests that protective adaptive T cell immunity following natural SARS-CoV-2 infection may persist for over 12 months, even without detectable humoral (antibody) responses. The study's introduction the results are clearly described and presented, and the discussion is appropriate. The abstract accurately represents the results and conclusions. Some comments have been provided as follows.
In the abstract, it is stated that 182 subjects participated in this study. However, the following sentence reports 84 females (46.2%) and 97 males (53.6%). There seems to be a potential inconsistency in the number and percentage calculation for the sex distribution. It may be advisable to verify this result for accuracy.
Could the author incorporate the sex, age, and median interval between the date of COVID-19 diagnosis and blood collection for each group in Table 1?
In lines 190-193, the calculated percentages for 148/182 should be 81.3%, and for 142/182, it should be 78.0%. Furthermore, the text mentions positive T cell reactivity against the S antigen in 148/182 cases, with 6/182 showing borderline S results and 27/182 being non-reactive. Similarly, for T cell responses against the N antigen, it is stated that 142/182 were positive, 9/182 had borderline answers, and 30/182 were non-reactive. However, the total count adds up to 181 out of 182 individuals. It appears that there may be one individual result missing. Could the author please review and confirm these results for accuracy?
Please add panels “a” and “b” to all figures. It would greatly assist the readers if the author could present Figure 1 in proportions of positive, borderline, and non-reactive results, making it easier to compare the favorable rates among the three groups rather than using the number of patients. Additionally, it would be beneficial if the author could provide more details in the Figure 1 legend to explain the distinctions between groups A, B, and C.
In lines 213-215, please mention the comparison result. For example, the results showed that median T cell reasons in group B were significantly elevated compared to groups A and C.
In Figure 3, could the author add the asterisk to represent the statistical significance?
In Figure 4-6, the text indicates “curve estimation,” but the results appear to exhibit a linear regression instead. Could the author add a more detailed explanation in the statistical analysis section? It would be helpful to clarify the meaning of "SR," if it stands for Spearman correlation, please explicitly state this in the statistical analysis.
In lines 237-246, it would be helpful if the author could rephrase the results, as there appears to be an inconsistency between the text and the actual results. The text mentioned “a weak positive correlation for the T cell response to the S antigen and negative correlation for the N antigen concerning time after infection (SR=0.187, P=0.17 for S, SR=-0.06, P=0.65 for N)”. However, the correlation coefficients show no significant difference. Therefore, it seems that there were no clear positive or negative trends observed between T cell responses to S and N antigens and time since infection."
Minor comments
In line 28, please change (37,9%) to 37.9%
Author Response
Comments and Suggestions for Authors:
This study focuses on assessing T cell immunity responses in individuals from the general population with a prior confirmed COVID-19 infection but no measurable levels of IgG antibodies. The research spans three time periods post-infection, and T-cell responses are evaluated using the T-SPOT.COVID test. The study suggests that protective adaptive T cell immunity following natural SARS-CoV-2 infection may persist for over 12 months, even without detectable humoral (antibody) responses. The study's introduction the results are clearly described and presented, and the discussion is appropriate. The abstract accurately represents the results and conclusions. Some comments have been provided as follows.
>> Thank you for positive consideration of our study.
In the abstract, it is stated that 182 subjects participated in this study. However, the following sentence reports 84 females (46.2%) and 97 males (53.6%). There seems to be a potential inconsistency in the number and percentage calculation for the sex distribution. It may be advisable to verify this result for accuracy.
>> Thank you for noticing it. It has been corrected accordingly - 85 females (46.7%) and 97 (53.3%) males. Please see line 39 in the revised manuscript.
Could the author incorporate the sex, age, and median interval between the date of COVID-19 diagnosis and blood collection for each group in Table 1?
>> We have added the demographic characteristics as suggested by the reviewer. Please see Table 1 in the revised manuscript.
In lines 190-193, the calculated percentages for 148/182 should be 81.3%, and for 142/182, it should be 78.0%. Furthermore, the text mentions positive T cell reactivity against the S antigen in 148/182 cases, with 6/182 showing borderline S results and 27/182 being non-reactive. Similarly, for T cell responses against the N antigen, it is stated that 142/182 were positive, 9/182 had borderline answers, and 30/182 were non-reactive. However, the total count adds up to 181 out of 182 individuals. It appears that there may be one individual result missing. Could the author please review and confirm these results for accuracy?
>> Thank you also for noticing it. We have added omicron instead of zero to our data sheet, therefore it was interpreted by the statistic method as a missing value. We have corrected the proportion by adding one case in non-reactive cases of both N and S antigens. Please see lines 208-211.
Please add panels “a” and “b” to all figures. It would greatly assist the readers if the author could present Figure 1 in proportions of positive, borderline, and non-reactive results, making it easier to compare the favorable rates among the three groups rather than using the number of patients. Additionally, it would be beneficial if the author could provide more details in the Figure 1 legend to explain the distinctions between groups A, B, and C.
>>We have made the corrections according to the suggestions of the reviewer. Please see Figures 1-7.
In lines 213-215, please mention the comparison result. For example, the results showed that median T cell reasons in group B were significantly elevated compared to groups A and C.
>>The correction has been made. Please see lines 226-227.
In Figure 3, could the author add the asterisk to represent the statistical significance?
>>The correction has been made. Please see Figure 3.
In Figure 4-6, the text indicates “curve estimation,” but the results appear to exhibit a linear regression instead. Could the author add a more detailed explanation in the statistical analysis section? It would be helpful to clarify the meaning of "SR," if it stands for Spearman correlation, please explicitly state this in the statistical analysis.
>> We have provided the necessary clarifications in terms of the type of statistics we used in each case in Statistical analysis section and also in results section and in the legends of the charts. Please see lines 180-184, 241, 243, 246, 248-249 and legends of figures 4-7.
In lines 237-246, it would be helpful if the author could rephrase the results, as there appears to be an inconsistency between the text and the actual results. The text mentioned “a weak positive correlation for the T cell response to the S antigen and negative correlation for the N antigen concerning time after infection (SR=0.187, P=0.17 for S, SR=-0.06, P=0.65 for N)”. However, the correlation coefficients show no significant difference. Therefore, it seems that there were no clear positive or negative trends observed between T cell responses to S and N antigens and time since infection."
>> We have made the appropriate improvement in text, please see lines 244-246.
Minor comments
In line 28, please change (37,9%) to 37.9%
>>The correction has been made. Please see line 41.
Reviewer 2 Report
Comments and Suggestions for Authors
The authors performed an epidemiological study to evaluate the levels of T cellular immunity response in a specific group of unvaccinated individuals of the general population with a prior confirmed COVID-19 infection and no measurable levels of IgG antibodies. Their data suggest that protective adaptive T cellular immunity following natural infection by SARS-CoV-2 may persist for over 12 months despite the undetectable humoral element. These results further enhance the evidence of long-term adaptive cellular immunity in unvaccinated COVID-19 convalescent individuals. I recommend it to be published with minor revisions described below.
1. Lines 188-193: According to what the authors described in section 2.2, there were 182 participants. However, in Figure 2, only 181 participants were included. Please explain the discrepancy.
2. Lines 226-229: In section 2.4, the authors claimed that no statistically significant correlations between the IFN-γ response to SAR-Cov-2 S and N antigen and time after infection were demonstrated…….demonstrating no changes in S and N antigen levels over time. It is unclear how the data are related to S and N antigen levels. Please explain. If it is a misstatement, please correct it.
3. Lines 42, 70, 132-149: Please correct the font size or the formatting.
4. Table 1: Please correct the formatting.
Author Response
Comments and Suggestions for Authors:
The authors performed an epidemiological study to evaluate the levels of T cellular immunity response in a specific group of unvaccinated individuals of the general population with a prior confirmed COVID-19 infection and no measurable levels of IgG antibodies. Their data suggest that protective adaptive T cellular immunity following natural infection by SARS-CoV-2 may persist for over 12 months despite the undetectable humoral element. These results further enhance the evidence of long-term adaptive cellular immunity in unvaccinated COVID-19 convalescent individuals. I recommend it to be published with minor revisions described below.
>> Thank you for positive consideration of our study.
- Lines 188-193: According to what the authors described in section 2.2, there were 182 participants. However, in Figure 2, only 181 participants were included. Please explain the discrepancy.
>> Thank you for noticing it. We have added omicron instead of zero to the statistical archive, therefore it was interpreted as a missing value. We have corrected the proportion by adding one case in non-reactive cases of both N and S antigens. Please see lines 208-211.
- Lines 226-229: In section 2.4, the authors claimed that no statistically significant correlations between the IFN-γ response to SAR-Cov-2 S and N antigen and time after infection were demonstrated…….demonstrating no changes in S and N antigen levels over time. It is unclear how the data are related to S and N antigen levels. Please explain. If it is a misstatement, please correct it.
>> We have revised the specific sentence for reader clarity. Please see lines 232-235.
- Lines 42, 70, 132-149: Please correct the font size or the formatting.
>>We have made the appropriate formatting. Please see lines 58-59, 86, 149-166.
- Table 1: Please correct the formatting.
>>We have made the appropriate formatting. Please see Table 1.